# E-Nose Technology for Mycotoxin Detection in Feed: Ready for a Real Context in Field Application or Still an Emerging Technology?

**DOI:** 10.3390/toxins15020146

**Published:** 2023-02-11

**Authors:** Federica Cheli, Matteo Ottoboni, Francesca Fumagalli, Sharon Mazzoleni, Luca Ferrari, Luciano Pinotti

**Affiliations:** 1Department of Veterinary Medicine and Animal Science, University of Milan, 26900 Lodi, Italy; 2CRC I-WE (Coordinating Research Centre: Innovation for Well-Being and Environment), University of Milan, 20100 Milan, Italy

**Keywords:** feed safety, mycotoxins, electronic nose

## Abstract

Mycotoxin risk in the feed supply chain poses a concern to animal and human health, economy, and international trade of agri-food commodities. Mycotoxin contamination in feed and food is unavoidable and unpredictable. Therefore, monitoring and control are the critical points. Effective and rapid methods for mycotoxin detection, at the levels set by the regulations, are needed for an efficient mycotoxin management. This review provides an overview of the use of the electronic nose (e-nose) as an effective tool for rapid mycotoxin detection and management of the mycotoxin risk at feed business level. E-nose has a high discrimination accuracy between non-contaminated and single-mycotoxin-contaminated grain. However, the predictive accuracy of e-nose is still limited and unsuitable for in-field application, where mycotoxin co-contamination occurs. Further research needs to be focused on the sensor materials, data analysis, pattern recognition systems, and a better understanding of the needs of the feed industry for a safety and quality management of the feed supply chain. A universal e-nose for mycotoxin detection is not realistic; a unique e-nose must be designed for each specific application. Robust and suitable e-nose method and advancements in signal processing algorithms must be validated for specific needs.

## 1. Introduction

Mycotoxins are one of the largest safety risks for the feed/food chain, with a negative impact on animal and human health, economy, and international trade of feed and food commodities [1,2,3,4,5,6]. Despite the availability of several strategies for prevention and control of fungal contamination, mycotoxin contamination in feed and food is unavoidable and unpredictable [6,7,8]. The challenge is to minimize the effects. The global trade of agricultural commodities, the climate change scenario, and the lack of harmonization in mycotoxin regulation are the main topics underlying the need of tools for the feed industry to manage the mycotoxin risk. It is undeniable that mycotoxin management demands an integrated approach using proactive, innovative, and improved strategic actions all along the feed chain [9]. Moving from science to practice, the first need is the availability of rapid and on-site analytical methods. At the feed industry level, notwithstanding the availability of advanced methods, there is a need for effective and rapid analytical methods for feed mycotoxin detection at the levels that are set by the regulations for an efficient mycotoxin risk management. Mycotoxins are regulated worldwide, but the set maximum levels vary greatly from country to country [10,11]. The European Union (EU) harmonized regulations on maximum levels of mycotoxins in feed among its member states [12,13,14]. A rapid, low-cost, high-throughput analytical approach for mycotoxin detection is a need at the industry level to make rapid management decisions on the acceptance or rejection of a lot [6]. The need for rapid methods and criteria to be considered for validation of methods to be used for mycotoxin detection were topics discussed in a “Special Issue: Rapid methods for mycotoxin detection” and “Special Issue: Rapid Detection of Mycotoxin Contamination” published in World Mycotoxin Journal and Toxins, respectively [15,16]. Within rapid methods, electronic nose (e-nose) may represent an attractive and promising method for mycotoxin detection.

After a brief survey on mycotoxin contamination in animal feed, this review provides an overview of the use of e-nose as an effective tool for rapid mycotoxin detection and management of the mycotoxin risk at feed business level.

## 2. Mycotoxin Contamination

Mycotoxins are secondary fungal metabolites. *Fusarium, Aspergillus, Penicillium*, and *Claviceps* spp. mycotoxins produced by represent the main contaminants of the feed supply chain, with important impact on animal health, productivity, and feed/food safety [5]. Of the more than 300 mycotoxins identified up to now, aflatoxins (AFs), aflatoxin B1 (AFB_1_), deoxynivalenol (DON), zearalenone (ZEA), fumonisins B_1_ and B_2_ (FBs, FB_1_, and FB_2_), ochratoxin A (OTA), T2, and H-T2 are regulated by EU legislation for animal feed [12,13,14]. 

Mycotoxin contamination occurs in feed all along the feed supply chain, including production, processing, storage, and distribution. Extensive surveys were carried out on mycotoxin occurrence in feed raw materials and complete feeds. However, forages must also be monitored because of their significant contribution to total mycotoxin intake [17]. Feed contamination may also represent a safety risk for humans because of the possible carry-over of mycotoxins into animal-derived food [18,19,20,21]. The main complete feed and feed raw materials analysed worldwide for mycotoxin contamination were grains and grain co-products (bran, corn gluten meal, dried distillers’ grains, and solubles). Less data are available for other feed ingredients, such as soybean meal, cotton seed, sorghum, cassava, peanut, and copra. [8,21,22,23,24,25,26,27,28,29,30,31,32,33,34,35].

Several important findings resulted from these multiannual mycotoxin surveys in animal feed. The overall results confirm that AFs, DON, FBs, OTA, T-2, and HT-2 toxins and ZEA are the main mycotoxins occurring in feed and are invariably found in cereal grains. Moreover, although the incidence of samples contaminated with mycotoxins above the EU legislative limit or recommended levels is low, there is a high variability, and several samples can exceed the levels. This confirms the need for a continual monitoring activity to check feed safety. Considerable differences in the mycotoxin profile (type and prevalence of mycotoxin contamination) in different geographic regions of the world and year by year variations have been reported [6,26,27,29,30,31]. Climatic and weather conditions (excessive moisture, temperature extremes, humidity, and drought) during critical plant growing stages, as well insect damage, crop systems, and some agronomic practices can cause plant stress and determine the severity of mycotoxin contamination [5,36,37,38]. In this scenario, climate change may have significant implications and effects on the distribution and occurrence of mycotoxins in the agri-food chain [39,40,41,42]. 

The second finding is that co-occurrence of mycotoxins is the norm not the exception. Multi-mycotoxin contamination was more prevalent in feed samples from Asia (82%) than from Europe and America (40%) [6]. The most frequently co-occurring mycotoxin combinations in compound feed were DON and ZEA; DON, T-2, and HT-2; ZEA, T-2, and HT-2; and DON, T-2, HT-2, and ZEA. Quite high co-occurrence level was found for OTA in combination with DON, T-2, and HT-2 [8,21,26,27,29,31,33,34].

Concerns about the safety of contaminated products have been further heightened by modified and emerging mycotoxin. As reported by several authors [43,44], the analysis of the mycotoxin content of samples containing these compounds can lead to their underestimation. The same author highlighted that such bias in masked mycotoxin detection might be due to several analytical issues. This implies that modified mycotoxins are hardly detected by routine analysis. This emerging issue was accessed by EFSA in a Scientific Opinion [45] on the risks for human and animal health related to the presence of modified forms of certain mycotoxins in food and feed. In the present opinion, all modified mycotoxins produced by plant or fungi metabolism, formed during feed/food processing, and resulting from the carry-over from contaminated feed are considered. Despite the increasing attention paid to modified mycotoxins, data on the formation, occurrence, toxicity, metabolic dynamics, and specific analytical methods are still rather scarce. Results from multiannual mycotoxin surveys in feed materials and complete feed indicate the presence of non-regulated mycotoxins: co-contamination of modified and emerging mycotoxins with regulated mycotoxins were reported [29,32]. Currently, worldwide legislation considers only mycotoxin mono-exposure data and does not address relevant mycotoxin co-contamination. Moreover, recently modified and emerging mycotoxins have been included in the EFSA risk assessments [46]. The impact of relevant mycotoxin combinations, regulated and not regulated mycotoxins, should be considered, and legislation must consider this topic in the near future.

## 3. Mycotoxin Analysis

The starting point of an effective mycotoxin analysis is sampling. This is a critical issue to obtain reliable results [47]. Research on this topic continues to evolve; however, sampling and sampling procedure are not the topic of this paper. For those who are interested, several papers are available for further and recent information [48,49,50,51,52]. Regarding sampling, recent publications on sampling techniques for grain dust and for pooling samples for mycotoxin screening could have a huge impact for the feed industry [53,54]. 

The official controls of feed and food are regulated by the Regulation (EU) 2017/625, Commission Regulation (EC) 152/2009, and Commission Regulation (EC) No 401/2006, laying down the methods of sampling and analysis for the official control of the levels of mycotoxins in feed and food, respectively [55,56,57]. These Regulations provide precise details regarding the methods of sampling, acceptance parameters, criteria for sample preparation, analytical performance criteria of the methods of analysis, and criteria for reporting and interpretation of the results. Identification criteria for mycotoxin limit of quantification (LOQ) have been the focus of a guidance document released by the European Commission [58]. At research levels, there is continual work to develop and validate methods for mycotoxin determination. Chromatography with MS/MS is the reference method for mycotoxin analysis in regulated matrices and is almost routinely performed. Studies regarding the implementation of LC-MS/MS methods, application of chromatography with targeted and non-targeted high-resolution mass spectrometry (HRMS), and optimisation of sample preparation for multi-mycotoxin analysis, including modified mycotoxins, have been reported in recent years [59,60]. 

At the feed industry level, the on-site quality and safety of products need to be continually monitored, and the adoption of a rapid, low-cost, high-throughput screening methods is a must for the management of mycotoxin risk [61]. Commercially available enzyme-linked immunosorbent assays (ELISA) kits are widely used due to their relatively low cost and easy application. ELISA assays meet the industrial needs in monitoring and surveillance programs as a “fit-for-purpose” tool. The development and validation of new ELISA and lateral flow immunoassays (LFIA) methods are still an area of great interest, including research on miniaturisation and multiplex new biosensors [59,60]. Among traditional method for mycotoxin detection, thin-layer chromatography (TLC) is considered to be an effective screening method for mycotoxins [62]. This traditional method has gained great significance as a simple, rapid, and economical method for quantitative detection, but the poor accuracy and low sensitivity make quantification difficult. This method is particularly effective in AFs and OTA determination [63].

In addition to conventional analytical methods, several authors recently evaluated electrochemical aptasensors for mycotoxin detection. Ong and co-workers [64] summarized, in a recent study, most recent advances in conventional methods and electrochemical aptasensors for mycotoxin detection. Considering this innovative technology, its main advantages are related to flexible modification of functional groups, high sensitivity, wide detection range of mycotoxin types due to its flexibility in electrode surface modification, simple operating procedure, and low cost of fabrication. The main disadvantage is the need for surface modification and signal amplification for a high sensitivity.

It is well known that fungal spoilage is responsible of organoleptic deterioration and off-flavour production associated with mycotoxins production [65,66]. Therefore, within rapid methods, e-nose, capable of recognizing simple or complex odours, could represent a fast and accurate tool in feed safety assessment by farmers and feed industry for mycotoxin screening.

## 4. Electronic Nose

An e-nose consists of an array of non-specific chemical sensors with partial specificity and an appropriate pattern-recognition system that can recognize simple or complex odours [67]. Sensors interact with different volatile organic compounds (VOCs), providing signals that can be utilized effectively as a unique flavour fingerprint of a product. The application of a robust pattern recognition system makes possible the identification and/or quantification of the odours [68,69]. The workflow of an e-nose analysis is reported in Figure 1.

There are several sensor devices for e-nose using different types of detection: optical, thermal, electrochemical, and gravimetric [70,71]. Within these types of sensors, the most popular e-nose sensors are metal–oxide semiconductor (MOS), metal–oxide semiconductor field-effect transistors (MOSFET), and conducting polymer (CP) and piezoelectric crystal sensors. The different sensor technologies affect their performance, such as response and recovery times, sensitivity, detection range, operating limitations, and inactivation by poisoning agents. Gas molecules interact with sensors by absorption, adsorption, or chemical reactions. According to the type of sensors, this reaction causes a modification of the sensor resistance, electrical conductivity, or resonance frequency, and these changes are measured as an electrical signal producing a fingerprint of VOCs. There was an instrumental evolution, leading to a wide diffusion of commercially available e-noses, automated, hybrid instruments with a combination of different sensor technologies, small size, and portable e-noses [72,73]. A universal e-nose, coping with every odour profile, is not realistic, and unique e-noses must be specifically designed and set up, and data processing must be validated for specific research work. Despite their different mechanisms, most of the sensors interact non-selectively with volatile molecules showing non-specific recognition. The result is a “fingerprint” of the VOCs. An instrumental evolution of e-nose is represented by a new generation of e-nose instruments based on ultra-fast gas chromatography. They share the fast-screening capability of other types of e-noses, while allowing, at the same time, specific identification and quantification of the detected volatile molecules. Applications of ultra-fast GC electronic nose are reported for food safety authentication and adulteration analysis [74,75,76]. 

Data analysis and pattern recognition (PARC) are fundamental parts of the e-nose analysis. E-nose analysis generates a great volume of data that requires the application of multivariate methods for data analysis. There are a variety of PARC methods that can be used depending on the type of data and the required results (Figure 2). For a comprehensive description and discussion regarding analysis of e-nose data, readers are referred to the literature [69,77,78,79]. 

Applications of e-nose analysis range from the feed/food industry and medical industry to environmental monitoring and process control [80,81,82,83]. The first applications of e-nose for food analysis date to the beginning of the 1990s [84,85]. At research and feed/food industry levels, e-nose technology has been employed for quality control of products: process, freshness, and maturity monitoring, shelf-life investigations, authenticity assessments, food fermentation process, animal source food, microbial pathogen, and pesticide detection [69,72,86,87,88,89]. From the first applications of the analysis with the e-nose, there have been no major changes in the application fields, while many differences can be found at the level of instrumental properties, data collection, and processing processes.

## 5. Volatilome: VOCs Associated with Fungal Metabolism

Volatile compounds are related to feed and food quality, aromatic attributes, and pleasant or unpleasant smell. Volatile compounds are a group of carbon-based chemicals with low molecular weight and high vapor pressure produced by bacteria and fungi as a result of their metabolism, and numerous of VOCs can originate from contamination in the field and during storage [90,91,92]. Volatile compounds can include alcohols, aldehydes, hydrocarbons, acids, ethers, esters, ketones, terpenes, furans, sulfur, and nitrogen-containing compounds. Fungi can produce similar VOCs, but the numbers and the amounts of individual VOCs vary. Differences found in the global pattern of VOCs are strictly correlated with fungi species and strains and growth conditions, such as substrate, nutrients, pH, humidity, and temperature. An on-line VOC database (http://bioinformatics.charite.de/mvoc/index.php?site=home) (accessed on 4 February 2023) reports more than 1000 VOCs from microorganisms; more than 300 of them are classified as fungal VOCs [91]. 

Several VOC markers differentiating grains were identified [93,94]. Volatile organic compound profile can be used as a fingerprint of different fungal species and toxigenic or non-toxigenic strains [93,95,96,97,98,99,100,101,102,103,104]. The main VOCs found in cultures of fungi grown on cereals belong to different categories, such as alcohols, aldehydes and ketones, benzene derivatives, hydrocarbons, and terpenes [66]. Magan and Evans (2000) conducted a milestone review of key studies carried out on the use of VOCs as potential indicators of fungal activity, giving evidence of relationships between the metabolic pathway leading to the formation of various VOCs and mycotoxin formation [66]. Since this review, a great number of new studies have been carried out to identify fungal VOCs in various cereal grains—grown under natural conditions or naturally infected. The most recent findings on VOCs in fungi contaminated grains are reported in Table 1.

Overall results indicate that (1) there is a wide range of fungal VOCs produced by spoilage fungi; (2) VOCs can be used as taxonomic markers of fungal species; (3) the presence of VOCs in naturally contaminated grain can be used as an early indicator of spoilage; and (4) more than single VOCs, the analysis of the VOC profile, by using multivariate analysis techniques, represents a powerful tool for the early detection and time evolution of fungal spoilage. 

Gas chromatography (GC)-based techniques have been used for the specific and sensitive analysis of VOCs and the volatilome profile. These techniques are reliable, specific, and sensitive, but expensive, time-consuming, and labor-intensive. In this scenario, according to the need of the feed industry, e-nose may represent a powerful tool for a rapid and on-site analysis of VOC profiles for identification of fungi contamination in agricultural commodities. Rapid analysis of mouldy and mycotoxin-contaminated agricultural commodities can reduce the risk of human/animal exposure to mycotoxins. Electronic nose was successfully used for VOC analysis and the early detection and differentiation between spoilage fungi and mycological quality grading of barley grains [65,114]. The study of Keshri and Magan [67] was the first one that showed that e-nose was able to differentiate between mycotoxigenic and non-mycotoxigenic strains of *Fusarium moniliforme* and *F. proliferatum* on the basis of their VOC production patterns [115]. From these studies, research on this topic has developed and increased. E-nose showed a very good discrimination capability for grain quality discrimination and detection of fungal contamination of cereal grain by discriminating contaminated and non-contaminated grains by *Penicillium* and *Fusarium* spp. and changes during the crucial stages of fungal growth [65,66,67,68,69,70,71,72,73,74,75,76,77,78,79,80,81,82,83,84,85,86,87,88,89,90,91,92,93,94,95,96,97,98,99,100,101,102,103,104,105,106,107,108,109,110,111,112,113,114,115,116,117,118,119,120,121,122,123]. 

By using an e-nose, the volatile compounds released by four *Fusarium* species were studied, and infected and non-infected wheat grains in the post-harvest chain were differentiated [113]. E-nose, combined with GC-MS, was able to identify the changes of volatile profile due to *Aspergillus* spp. growth in rice kernels [122]. Visualization of VOCs profiles of *Aspergillus oryzae* contaminated brown rice was possible and useful for early detection of fungal infection [106]. A systematic review on detection and identification of fungal species by e-nose technology in various fields of application beyond that of food safety has been recently published [83]. 

In addition to research on fungal VOCs as indirect indicators of fungal growth, in recent years, studies on fungal VOC production explored new topics: role in ecosystems (many ecological interactions among fungi and plants, arthropods, bacteria, and other fungi are mediated by VOCs), development of environmentally friendly biopesticides, and use in biotechnological applications (biofuel, biocontrol, and mycofumigation) [10,123,124,125]. Moreover, there is increasing experimental evidence that some fungal VOCs may be toxic. Bennett and Inamdar (2015) proposed the term “volatoxin” to describe VOCs with toxigenic properties [89].

## 6. E-Nose for Mycotoxin Detection

Rapid evaluation of feed quality and safety represents a challenge for the feed industry for mycotoxin risk management. As previously discussed, the potential for using sensor arrays to discriminate between toxigenic and non-toxigenic fungi exists [96]. Detection of mycotoxin contamination by e-nose is based on the detection of changes in the composition of VOCs produced by mycotoxigenic fungi during their growth and biochemical processes. Terpene production has been found to correlated to the production of AFB_1_ [104]. The volatile terpene trichodiene is the first metabolite in the trichothecene biosynthesis pathway [126]. The production of volatile terpenes relates to the formation of *Fusarium* trichothecene mycotoxins [127,128,129]. Volatile sesquiterpene hydrocarbon has been found to be a marker for *Penicillium roqueforti* strains, producing PR toxin [130]. No volatile compound uniquely related to OTA formation has been found [131]. The content of DON in durum wheat has been found to be positively (trichodiene, longifolene, 3-methyl butanal, tridecane, g-caprolactone, and 6,10,14-trimethyl-2-pentadecanone) and negatively (hexadecane, 2,3,7-trimethyl-decane, and 4,6-dimethyl-dodecane) correlated to the pattern of VOCs [112]. 

Recent applications reporting specific applications of e-nose for mycotoxin detection are reported in Table 2.

E-nose can be a powerful tool for quantitative/semiquantitative prediction of mycotoxin levels in grains. To move e-nose analysis from research to the industrial level, there are several questions that need answers. The main points that must be considered are: (1) the presence of maximum levels for mycotoxins in feed for practical enabling of rapid decision-making regarding the acceptance or rejection of lots of cereal and ensuring safety standards; (2) mycotoxin co-contamination; and (3) the classification and prediction accuracy of the e-nose-based model. 

Regarding the first point, e-nose was able to predict the FB content of maize cultures for high and low contamination levels [138]. E-nose was able to discriminate DON-contaminated and non-contaminated wheat and aflatoxin-contaminated and non-contaminated maize [139,141]. E-nose analysis was able to discriminate durum wheat samples at contamination levels close to that of the DON maximum limit set by the EU regulations [111,136]. E-nose was able to detect OTA and to predict whether the OTA level was below or above 5 ug/kg, representing the maximum level for OTA in cereals for food by EU regulations [140]. 

Mycotoxin co-contamination is the rule. E-nose analysis has been proposed to detect aflatoxin and fumonisin co-contamination in maize [134]. E-nose was effective in detecting co-contaminated samples, but with a low classification accuracy of 61% and 67%, respectively, of samples correctly classified for co-contamination using LDA. 

The type and percentage of misclassified samples are important and are critical issues in determining the performance and accuracy of e-nose analysis. Olsson et al. (2002) investigated the possibility of using fungal VOCs as indicators of two mycotoxins (OTA and DON) in barley, using both e-nose and GC/MSD [140]. In that study, the authors reported that the e-nose misclassified less than 20% of samples in the case of OTA. The DON level could be estimated using a partial least square (PLS) model constructed using the sensor signals from the e-nose. The detection of co-contamination was not the aim of this study, although several samples were found to be contaminated by both OTA and DON. Overall results indicate that, for single mycotoxin contamination, high discrimination accuracy between contaminated and non-contaminated grain has been reported. However, when mycotoxin co-contamination occurs, the predictive accuracy of e-nose is still limited and unsuitable for industrial applications in a real context. 

Finally, several e-noses are available on the market, and they can be customised according to the need. The very recent study of Machungo et al. (2022) compared the performance of three e-nose instruments for the detection of VOCs in maize contaminated with aflatoxins (Table 2) [132]. One of the three tested instruments (DiagNose) was more effective than the other two (Fox 3000 and Cyranose) for the detection of aflatoxin contamination of maize, with a cross-validated classification accuracy for the different sample classes ranging from 81% to 94%. 

## 7. Conclusions

E-nose represents a powerful tool in the feed chain for quality and safety control and monitoring. E-nose offers potential as a rapid and cost-effective diagnostic tool for mycotoxin contamination screening at the market entry level. However, before e-nose laboratory-based assays can move from research into the feed industry and become a reality, we must face and overcome several challenges to improve e-nose performance.

The future challenges are: the sensor materials, data analysis, pattern recognition systems, and a better understanding of the industrial needs related to safety and quality control of the feed supply chain. A universal e-nose for mycotoxin detection is not realistic; a unique e-nose must be designed for each specific application. Limitations still exist regarding sensitivity and selectivity of sensors. The major drawback is represented by sensors’ sensitivity to environmental conditions, particularly humidity and temperature. Improved modelling, correlation between chemical markers and sensor responses, and robust and suitable e-nose methods and advancements in signal processing algorithms must be validated for specific needs. In the field of mycotoxin co-contamination detection, the predictive accuracy of the e-nose models is still limited for industrial applications in a real context. Last, but not least, specific sampling models must be carefully selected to enhance the accuracy of e-nose analysis.

Appendix A could be found in Appendix A.

## Figures and Tables

**Figure 1 toxins-15-00146-f001:**
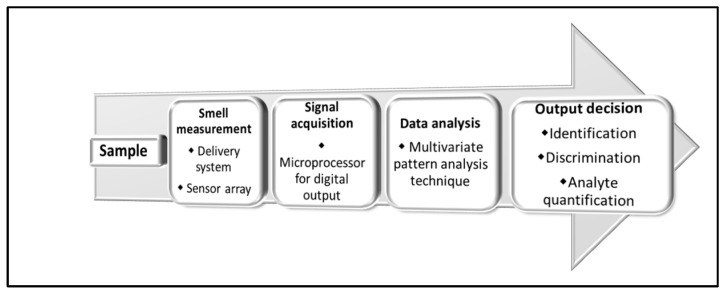
Analytical workflow for e-nose analysis.

**Figure 2 toxins-15-00146-f002:**
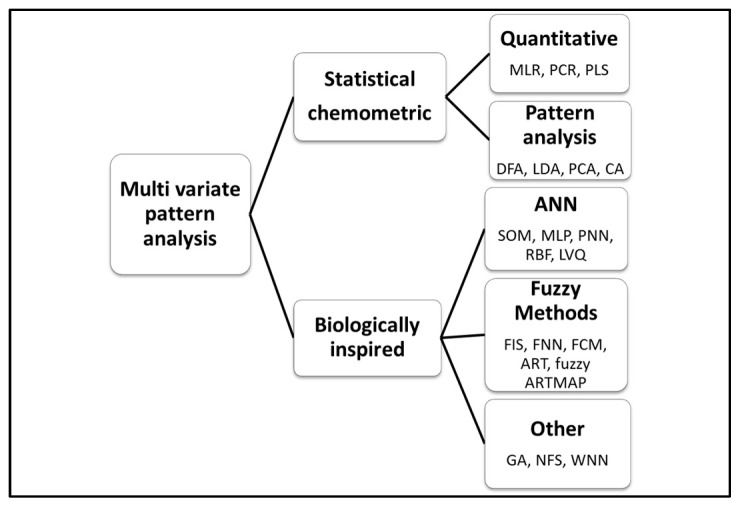
Overview of the most commonly used multivariate pattern analysis methods of e-nose data.

**Table 1 toxins-15-00146-t001:** VOC profiling due to fungal contamination in cereals: most recent acquisitions (not exhaustive list).

Samples	Fungal Contamination (*/**)	VOC Analysis	VOCs	References
Maize	*Aspergillus flavus* (*)	GC-IMS	A total of 55 VOCs were identified. Ethyl acetate-D and 3-hydroxybutan-2-one-D are potential biomarkers specific to *A. flavus* contamination. Aflatoxin B_1_ is positively correlated with the level of (E)-2-octenal-M, benzene acetaldehyde, (E)-hept-2-enal-M, 2- heptanone-D, and 2-pentyl furan.	[105]
Jasmine brown rice	*Aspergillus oryzae* (*)	SPME/GC-MSD	A total of 11 VOCs were identified. Octane, 2,2,4,6,6-pentamethylheptane, decane, dodecane, toluene, ethanol, 1-pentanol, 1-hexanol, 1-octen-3-ol, 2-heptanone, and 2-pentylfuran could be used as volatile markers for *A. oryzae* contamination.	[106]
Rice	*Aspergillus* strains (*A. candidus, A. fumigatus,* and *A. clavatus*) (*)	HS-GC-MSD	A total of 25 VOCs were identified. Decanal, 1-octanol, 1-tridecanol, nonanal, diethyl phthalate, α-cedrene, cyclododecene, and cis-thujopsene can be considered as markers of infected rice samples, with changes during the storage period.	[107]
Wheat	Ten fungal species, *Alternaria* (4), *Cladosporium* (3), *Penicillium* (2), *Aureobasidium* (1), and *Fusarium graminearum* (1) (*)	GC-FID, GC-MSD	A total of 57 VOCs were identified. Cyclooctasiloxane and hexadecamethyl combination and pentadecane can be considered as markers of early detection of postharvest fungi in grain for *A. alternata* and *A. infectori*, respectively. Naphthalene was identified only in the headspace of *C. herbarum*	[100]
Hybrid and dwarf maize	*Fusarium graminearum* and *F. verticillioides* (*)	SPME/GC-MSD	A total of 23 VOCs were identified (12 from dwarf and 15 from hybrid maize). Both varieties shared six common markers: (+)-longifolene, β-farnesene, β-macrocarpene, and trichodiene. Qualitative variability in VOCs was observed upon infection of different *Fusarium* species: trichodiene was detected only from *F. graminearum.*	[108]
Barley (malting procedure)	*Fusarium poae* (*)	SPME/GC-MSD	A total of 46 VOCs. Volatile aldehyde fractions were influenced by *F. poae* contamination during malting.	[109]
Maize	*Fusarium graminearum, F. verticillioides,* and *F. subglutinans*	SPME/GC-MSD OLS/GC-MSD	A total of 22 VOCs were identified. 3-hexen-1-ol, heptan-2-ol, 1-octen-3-ol, octan-3-one, octan-3-ol, β-selinene, α-selinene, β-macrocarpene, and β-bisabolene: markers for the early detection of *Fusarium* infection.	[110]
Durum wheat	*Fusarium poae* (*)	SHS-SPME/GC-MSD	A total of 29 VOCs were identified. Levels of ethyl acetate, ethanol, 3-methylbutanol ethyl decanoate, ethyl decenoate, 2-phenylethyl acetate, 3-methylbutanal, hexanal, phenylethyl alcohol, 3-hydroxy-2-butanone, and acetic acid changed as a function of time after inoculation.	[111]
Durum wheat	(**) DON < 1000 mg/kg; 1000 mg/kg ≤ DON ≤ 2500 mg/kg; DON > 2500 mg/kg.	HS-SPME/GC-MSD	A total of 70 VOCs were identified. Trichodiene, longifolene, 3-methyl butanal, tridecane, g-caprolactone, and 6,10,14-trimethyl-2-pentadecanone: positively associated with DON; Hexadecane, 2,3,7-trimethyl-decane, and 4,6-dimethyl-dodecane: negatively associated with DON	[112]
Barley, Oats, and rye	(**) analysis for trichothecenes A and B	GC/MSD	A total of 46 VOCs were identified. The most significant VOCs to differentiate infected from non-infected cereals: [E, E]-3,5 octadien 2-one, 1-heptanol, naphthalene, p-xylene and dimethyl sulphone, and trichodiene.	[100]
Soft wheat	*Fusarium graminearum, F. culmorum, F. cerealis,* and *F. redolens* (*)	SPME/GC-MSD	A total of 16 VOCs were identified. 2-methyl-1-propanol, 3-methylbutanol, 1-octen-3-ol, and 3-octanone were infection-specific.	[113]

*: artificially inoculated; **: naturally contaminated; GC-IMS: Gas Chromatography–Ion Mobility Spectrometry; SPME/GC-MSD: Solid-phase Microextraction/Gas Chromatography–Mass Spectrometry; HS-GC-MSD: Headspace-Gas Chromatography–Mass Spectrometry; GC-FID: Gas Chromatography–Flame-ionization detection; GC-MSD: Gas Chromatography–Mass Spectrometry; OLS/GC-MSD: Open-loop stripping/Gas Chromatography–Mass Spectrometry; SHS-SPME/GC-MSD: Static headspace–Solid-phase Microextraction/Gas Chromatography–Mass Spectrometry; HS-SPME/GC-MSD: Headspace–Solid-phase Microextraction/Gas Chromatography–Mass Spectrometry.

**Table 2 toxins-15-00146-t002:** Application of e-nose for mycotoxin detection in cereals.

Mycotoxins	Sample (*/**)	E-Nose/Sensor Array	Data Analysis	Tested Hypothesis	References
AFs	Maize (*)	Fox 3000/(6 SnO_2_ and 6 CTO); Cyranose 320; and DiagNose/12 MOS	SVM, k-NN	Aflatoxins—two classes: below and above 10 µg/kg (ppb)	[132]
DON	Wheat (**)	AIR PEN 3/10 MOS	CART	Discrimination among four DON contamination thresholds: 1750, 1250, 750, and 500 µg/kg	[133]
AFB_1,_ FUM	Maize (**)	AIR PEN 3/10 MOS	ANN, LR¸ DA	Discrimination at levels above or below the legal EU limits #	[134]
Afs, FBs	Maize (**)	AIR PEN3/10 MOS	DFA	Three classes of contamination: below the EU regulatory limits ##, single-contaminated, and co-contaminated	[135]
DON	Wheat bran (**)	AOS-ISE Nose 2000/12 MOS	DFA	Two contamination classes: A: DON ≤ 400 µg/kg and B: DON > 400 µg/kg	[136]
DON	Durum wheat (**)	AOS- ISE Nose 2000/12 MOS	DFA	Three contamination classes: A: DON ≤ 1000 mg/kg; B: 1000 < DON ≤ 2500 mg/kg; and C: DON > 2500 mg/kg.	[112]
DON	Durum wheat (**)	AIR-PEN2/10 MOS	PCA, CART	Three clusters based on the DON content proposed by the European legislation: A: non-contaminated; B: contaminated below the limit (DON ≤ 1750 μg/kg); and C: contaminated above the limit (DON > 1750 μg/kg)	[137]
FBs	Maize (*)	EOS^835^/6 MOX	PCA, PLS	FBs: low content below 1.6 mg/kg (average 1.0 mg/kg) vs. high content above1000 mg/kg	[138]
AFs	Maize (**)	AIRSENSE PEN2/10 MOS	PCA, LDA	Aflatoxin-containing samples and aflatoxin-free samples	[139]
OTA, citrinin	Durum wheat (**)	FOX 3000, Alpha-MOS/12 sensors	CORR	OTA, citrinin time changing during storage (25 weeks)	[140]
DON	Durum wheat (**)	PEN2/10 MOS	PCA, MR	DON-containing samples and DON-free samples	[141]
DON, OTA	Barley (**)	VCM 422/10 MOSFET, 6 SnO_2_, and 1 Gascard CO_2_	PCA, PLS	The OTA level varied between 0 and 934 mg/kg; the DON content varied between 0 and 857 mg/kg	[142]

AFs: aflatoxins; DON: deoxynivalenol; FBs: fumonisins; OTA: ochratoxin; DFA: discriminant function analysis; PCA: principal component analysis; CART: classification and regression tree analysis; PLS: partial least squares analysis; LDA: linear discriminant analysis; CORR: correlation analysis; SIMCA: soft independent modelling of class analogy; * artificially inoculated; ** naturally contaminated; SnO_2_: oxide sensors; CTO: chromium titanium oxide sensors; MOS: metal–oxide sensors; MOSFET: metal–oxide semi-conductor field-effect transistor sensors; SVM: support vector machine; k-NN: k-nearest neighbour; ANN: artificial neural network; LR: logistic regression; DA: discriminant analysis; DFA: discriminant function analysis; MR: multiple regression analysis; # AFB_1_: 5 μg/kg, FBs: 4000 μg/kg; and ##: AFs < 5 ppb, FBs FM < 4 ppm.

## Data Availability

Not applicable.

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
