# Peer review of "E-Nose Technology for Mycotoxin Detection in Feed: Ready for a Real Context in Field Application or Still an Emerging Technology?"

_toxins, 2023, doi:10.3390/toxins15020146_

Round 1

Reviewer 1 Report

The authors review the application of the e-nose technology to detect mycotoxins in feed. Although it is a good emerging technology, it is clear that it still needs quite a lot of improvement and it is not ready to be used as 'the technology' for mycotoxin detection either in food or feed (or in any other applications). This review lacks novelty, as many other articles have been published on this issue. Is there any other way that the authors could add more value to the review?

I overall missed the chemical structure of (at least) the most relevant mycotoxins discussed in the review, as well as more examples of mycotoxin detection by e-nose. Also, a table with LOD/LOQ in several feed would be appreciated. Finally, a figure of how an E-nose looks like would be good to add.

The review is well written, only a few minor comments:

key contributions: E-nose represents

Line 57: produced by fungal species. Redundant, please remove.

Line 80: need of - need for

Line 359: E-nose represents

Author Response

Thank you to the reviewer for all the valuable comments. Point by point answers to the comments are reported below.

The authors review the application of the e-nose technology to detect mycotoxins in feed. Although it is a good emerging technology, it is clear that it still needs quite a lot of improvement and it is not ready to be used as 'the technology' for mycotoxin detection either in food or feed (or in any other applications). This review lacks novelty, as many other articles have been published on this issue. Is there any other way that the authors could add more value to the review?

AU: Thanks to the comments of all the three reviewers, the MS was implemented in order to improve the scientific value. Several elements of novelty were included and discussed, such as providing more information on the review of e-nose technology for mycotoxin detection itself, providing a comparison with other analytical method, give more information on detection of masked mycotoxins in feeds. Furtherly, a brief description of some other detection methods, such as TLC and electrochemical detection, was included.

I overall missed the chemical structure of (at least) the most relevant mycotoxins discussed in the review, as well as more examples of mycotoxin detection by e-nose.

AU: The  chemical structure of  the most relevant mycotoxins discussed in the review was added as supplementary material. As written in the text of the manuscript, applications of e-nose analysis are well documented in the field of quality control of products: process, freshness and maturity monitoring, shelf-life investigations, authenticity assessments, food fermentation process, animal source food, microbial pathogen, and pesticide detection. In this paper we focused on e-nose application for mycotoxin detection in the main ingredients of feed, and therefore critical for the feed industry.

Also, a table with LOD/LOQ in several feed would be appreciated.

AU: Actually, authors believe that a table including LOD/LOQ in several feed analysis might be misleading for the readers. Indeed, as mentioned in the MS, the use of e-nose in the case of single mycotoxin contamination, high discrimination accuracy between contaminated and non-contaminated grain has been reported. A robust and suitable e-nose method has been validated to discriminate wheat samples at DON contamination levels close to the maximum permitted limit set by the European Union. However, for mycotoxin co-contamination further researches are needed. To date, in the field of mycotoxin co-contamination detection, the predictive accuracy of the e-nose models is still limited and unsuitable for industrial applications in a real context. From a technical perspective, contaminated samples misclassified as non-contaminated represent the worst outcome under in-field conditions in selecting samples that must undergo further accurate and quantitative analysis. For the reasons reported above, authors believe that including a table reporting LOD/LOQ might overestimate the potential of e-nose.

Finally, a figure of how an E-nose looks like would be good to add.

AU: there are several e-noses commercially available. Therefore, one picture could be reductive, and we preferred to describe the analytical workflow for the e-nose analysis. In the text, references [69,70], a description of commercially available e-noses are reported. Interested readers can refer to these citations.

The review is well written, only a few minor comments:

key contributions: E-nose represents

Line 57: produced by fungal species. Redundant, please remove.

Line 80: need of - need for

Line 359: E-nose represents

AU: changed were made in the text 

Reviewer 2 Report

This article intended to provide an overview on E-Nose technology for mycotoxins detection in feed. It is interesting, but the major disadvantage is until now, there is lack of information on this part. In most part of this review, the descriptions are not related to the detection of mycotoxins by E-nose. From this article, we could not find enough information on the detection of mycotoxins by e-nose. Some comments:

Abstract: Please provide more information on the review of e-nose technology for mycotoxin detection itself, not only the disadvantages and limitations.

What about the detection of masked mycotoxins in feeds, such as D3G, which has also been found in feeds?

For the detection of mycotoxins (Section 3), some other detection methods, such as TLC, electrochemical detection, should also be given a brief introduction.

    The e-nose technology is frequently used for the detection of volatile organic compounds (VOCs). What about the mycotoxins?

Section 5, most of this part, including the table, described the methods used for VOCs, which were not related to mycotoxins.

Section6, it is not the E-nose detection for mycotoxin.  

Author Response

Thank you to the reviewer for all the valuable comments. Point by point answers to the comments are reported below.

This article intended to provide an overview on E-Nose technology for mycotoxins detection in feed. It is interesting, but the major disadvantage is until now, there is lack of information on this part. In most part of this review, the descriptions are not related to the detection of mycotoxins by E-nose. From this article, we could not find enough information on the detection of mycotoxins by e-nose.

AU: Very good comment, thank you. Applications of e-nose analysis are well documented in the field of quality control of products: process, freshness and maturity monitoring, shelf-life investigations, authenticity assessments, food fermentation process, animal source food, microbial pathogen, and pesticide detection. Specific applications of e-nose for mycotoxin detection are not many, as reported in the manuscript with a focus on feed ingredients. However, e-nose could represent a powerful rapid tool for mycotoxin detection. Future research on sensor materials and data collection and analysis could allow to overcome the still existing limits. Regarding the reviewer comment “In most part of this review, the descriptions are not related to the detection of mycotoxins by e-nose”, volatilome analysis by e-nose is related to mycotoxin analysis, as VOCs are potential indicators of fungal activity, and there is evidence of relationships between the metabolic pathway leading to VOCs’ and mycotoxin formation. Differences found in the global pattern of VOCs are strictly correlated with fungi species, strains, and mycotoxigenic and non-mycotoxigenic fungi. Fungal volatile metabolites can be used as an indicator of mycotoxins occurrence in feed/food products, and relevant patterns of VOCs positively or negatively associated with mycotoxins have been found, as discussed in the text.

Abstract: Please provide more information on the review of e-nose technology for mycotoxin detection itself, not only the disadvantages and limitations.

AU: the abstract was modified to give more information regarding e-nose technology for mycotoxin detection itself (highlighted in red in the text).

What about the detection of masked mycotoxins in feeds, such as D3G, which has also been found in feeds?

AU: the reviewer highlighted a good point. Up to now, at the best of my knowledge, e-nose analysis was applied to detect regulated mycotoxins. However, in order to implement the MS a brief section describing the detection of masked mycotoxins in feedswas included in the text, section 2 (highlighted in red in the text).

For the detection of mycotoxins (Section 3), some other detection methods, such as TLC, electrochemical detection, should also be given a brief introduction.

AU: in order to implement the MS, a brief section describing other detection methods, such as TLC and electrochemical detection, was included in section 3 (highlighted in red in the text). 

The e-nose technology is frequently used for the detection of volatile organic compounds (VOCs). What about the mycotoxins?

Section 5, most of this part, including the table, described the methods used for VOCs, which were not related to mycotoxins.

AU: As previously stated, volatilome analysis by e-nose is related to mycotoxin analysis, as VOCs are potential indicators of fungal activity, and there is evidence of relationships between the metabolic pathway leading to VOCs’ and mycotoxin formation. Differences found in the global pattern of VOCs are strictly correlated with fungi species, strains, and mycotoxigenic and non-mycotoxigenic fungi. Fungal volatile metabolites can be used as an indicator of mycotoxins occurrence in feed/food products, and relevant patterns of VOCs positively or negatively associated with mycotoxins have been found, as discussed in the text.

Section 6, it is not the E-nose detection for mycotoxin.  

AU: Actually, section 6 is a review of the available published paper on recent applications of e-nose for mycotoxin detection in feed. In table 2, data on Mycotoxins, Analysed samples, E-nose/sensor array, Data analysis, and Tested hypothesis are reported in details.

Reviewer 3 Report

The paper is written very clearly and is easy to follow, but you need to make some corrections.... 

Author Response

Thank you to the reviewer for all the valuable comments. Point by point answers to the comments are reported below.

The paper is written very clearly and is easy to follow, but you need to make some corrections....  peer-review-26017120.v1.docx

AU: Thank you for your comments.

Corrections have been included in the text of the revised manuscript

Recently, the LOQ, is the dominant item, while the LOD is not so important.

AU: LOD has been deleted from the text

There are several scientific papers regarding the use of e-noses for the pesticide residues analysis. Maybe it's good to mention that.

AU: Scientific papers regarding the use of e-noses for the pesticide residues analysis were included in the text.

Round 2

Reviewer 1 Report

The authors successfully addressed all the comments and, thus, paper is now ready for publication.